# Lamb Wave Detection for Structural Health Monitoring Using a ϕ-OTDR System [note 1]

**DOI:** 10.3390/s22165962

**Published:** 2022-08-09

**Authors:** Rizwan Zahoor, Enis Cerri, Raffaele Vallifuoco, Luigi Zeni, Alessandro De Luca, Francesco Caputo, Aldo Minardo

**Affiliations:** Department of Engineering, University of Campania “Luigi Vanvitelli”, Via Roma 29, 81031 Aversa, Italy

**Keywords:** optical fibers, distributed sensing, structural health monitoring

## Abstract

In this paper, the use of a phase-sensitive optical time-domain reflectometry (ϕ-OTDR) sensor for the detection of the Lamb waves excited by a piezoelectric transducer in an aluminum plate, is investigated. The system is shown to detect and resolve the Lamb wave in distinct regions of the plate, opening the possibility of realizing structural health monitoring (SHM) and damage detection using a single optical fiber attached to the structure. The system also reveals the variations in the Lamb wave resulting from a change in the load conditions of the plate. The same optical fiber used to detect the Lamb waves has also been employed to realize distributed strain measurements using a Brillouin scattering system. The method can be potentially used to replace conventional SHM sensors such as strain gauges and PZT transducers, with the advantage of offering several sensing points using a single fiber.

## 1. Introduction

The use of sensors for the diagnosis and prognosis of a structure during the operating life is increasingly attracting attention from both the research community and industry given their benefits in terms of maintenance, repair operations, and improved design practice. In particular, structural health monitoring (SHM) is of high interest as it provides automated diagnosis of structural components. Among the various SHM techniques, those based on Lamb waves are widely acknowledged as one of the most encouraging tools for quantitative identification of damage in metallic and composite structures [1,2,3]. Lamb waves are guided ultrasound waves that can be excited in thin-walled structures. Depending on their frequency, they can propagate over long distances, making them ideal for long-range inspection purposes [4]. Lamb wave-based methods may provide an indication of the occurrence of damage, an assessment of its position, and a quantitative estimation of its severity; all this information can significantly contribute to the prediction of the residual service life of the component [5].

While the excitation and detection of Lamb waves are usually achieved with piezoelectric (PZT) transducers, fiber Bragg grating (FBGs) optical sensors have also been proposed for Lamb wave detection [6,7,8,9,10]. FBG sensors have the advantage of being immune to electromagnetic interference. Moreover, they are small, lightweight, and capable of working in unfavorable environments such as in wet areas, underwater, and at high temperatures. On the downside, several FBG sensors, positioned in different directions, must usually be deployed to detect the Lamb waves at different points of the investigated structure [11].

In this paper, we investigate the use of a distributed acoustic sensor based on phase-sensitive optical time-domain reflectometry (ϕ-OTDR) for Lamb wave detection in an aluminum plate, opening a new perspective for Lamb-wave-based SHM. A laboratory setup implementing the ϕ-OTDR technique has been used to detect the Lamb waves in distinct regions of the plate. The same optical fiber has also been used to measure the strain distribution through stimulated Brillouin scattering. The possibility to use a single optical fiber for multiparametric sensing is especially important, as the temperature and strain conditions of the structure under investigation influence the Lamb waves propagating into the structure [12,13]. Therefore, measuring these quantities may provide a means to compensate the changes in the detected Lamb waves and distinguish them from the changes induced by damage.

## 2. Phase-Sensitive Optical Time-Domain Reflectometry

The phase-sensitive time-domain reflectometry (ϕ-OTDR) exploits the Rayleigh scattering in an optical fiber to detect any acoustic disturbance acting on the fiber itself [14]. The method is usually achieved by injecting a highly coherent optical pulse from one end of the fiber. The resulting backscatter signal due to Rayleigh scattering is then recorded as a function of time. The backscatter signal is the result of the interference of multiple backscattered contributions within the pulse length; therefore, any acoustic events occurring along some portion of the fiber will change the relative distribution of the scatterers within that portion, and therefore, the amplitude of the locally backscattered signal. The perturbation also induces a phase delay, so the phase of the backscattered light changes from the perturbation section onwards. The local phase change, and therefore, the applied perturbation, can be recovered by calculating the differential phase between nearby positions.

In this work, the ϕ-OTDR method has been implemented following the scheme shown in Figure 1. In brief, the output of a narrowband (<7 Hz) laser, operating at 1550 nm, is first divided into two paths using a 90/10 optical coupler: the optical local oscillator (OLO) and the probe. In the probe branch, an acoustic optical modulator (AOM) is used to carve optical pulses with a duration of 20-ns while also prompting a shift of 300 MHz on the carrier frequency. After being amplified by an erbium-doped fiber amplifier, the probe pulses are launched into the sensing fiber through an optical circulator (OC). The Rayleigh backscattered light is mixed with the OLO through a 3 dB optical coupler whose output ports are connected to a balanced photodetector.

The mixing process generates a beat signal at an interference frequency (IF) equal to the frequency shift added through the AOM (300 MHz), which is digitized with a sampling rate of 2 GS/s. The acquired waveforms are exported and processed offline using a MATLAB script which performs I/Q demodulation and phase unwrapping of the differential phase, i.e., of the phase difference between sampling positions of the fiber located at a distance equal to the spatial resolution (2 m in our tests).

The processing of the acquired waveforms can be described as follows: The signal and the local oscillator sent to the 2 × 2 optical coupler can be expressed as [15]:(1)$$E_{s}\left(t\right)=A_{S}\left(t\right)\exp\left[j\left(2\pi \left(f+\Delta f\right)t+\phi \left(t\right)+\phi_{1}\right)\right]$$
(2)$$E_{LO}\left(t\right)=A_{LO}exp\left[j\left(2\pi ft+\phi_{2}\right)\right]$$
where $A_{S}\left(t\right)$ and $A_{LO}$ are the amplitudes of the signal light and local light, respectively, f is the laser frequency, Δf is the frequency shift imposed by the AOM, ϕ(t) is the phase change of the signal induced by the external perturbation, and $\phi_{1}$ and $\phi_{2}$ are the initial phases of the signal and OLO, respectively. At the receiver end, the mixing of the signal and local lights produces a signal at the output of the balanced photodetector proportional to the optical power:(3)$$P_{BPD}\propto 2A_{S}\left(t\right)cos\theta \left(t\right)A_{LO}\cos\left[2\pi \Delta ft+\phi \left(t\right)+\phi_{1}-\phi_{2}\right]$$
where θ(t) considers the polarization mismatch between local and signal lights which also temporally varies. The beat signal recorded by the acquisition card is processed using the I/Q demodulation method [16], according to which the in-phase and quadrature signals can be obtained by first multiplying the detected signal by cos(2πΔft) and sin(2πΔft), respectively, and then applying a low-pass filter to remove the high-frequency components. This provides two quadrature components of the signal:(4)$$\{\begin{array}{c} I\left(t\right)\propto A_{S}\left(t\right)A_{LO}sin\left(\phi \left(t\right)\right)\\ Q\left(t\right)\propto A_{S}\left(t\right)A_{LO}cos\left(\phi \left(t\right)\right) \end{array}$$

Finally, the amplitude $A_{S}\left(t\right)$ and phase ϕ(t) of the Rayleigh backscattered light can be calculated as [17]:(5)$$\{\begin{array}{c} A_{S}\left(t\right)\propto \sqrt{I^{2}+Q^{2}}\\ \phi \left(t\right)=\tan^{-1}\left(Q/I\right)+2k\pi \end{array}$$
where k is an integer. Therefore, the phase information ϕ(t) can be accurately demodulated by (5). In Equation (5), the time co-ordinate t is converted into a spatial coordinate z using $z=\frac{c}{2n}t$, where c is the light velocity in the vacuum and n is the refractive index of the fiber. The final processing stage consists of calculating the differential phase which is directly proportional to the applied strain, i.e., Δϕ(z)=ϕ(z+GL)−ϕ(z), with GL = 2 m.

## 3. Experimental Results

The tests reported in this section have been carried out using the setup shown in Figure 1 and a specialty sensing fiber (AcoustiSens by OFS). This fiber uses a distributed weak Bragg grating inscribed along its length in order to enhance the backscattered Rayleigh signal by 13 dB compared to a conventional single-mode fiber. The optical losses are about 0.3 dB/km. The aluminum plate selected for the experiment has dimensions of 500 mm × 500 mm × 2 mm and was kept either in rest conditions or suspended using two pillars as schematically shown in Figure 2. The Lamb waves were generated into the metallic plate using the piezo-ceramic actuator disc PZT1 located at the center of the plate. The piezoelectric disc had a diameter of 10 mm and a thickness of 0.25 mm. Four additional piezoelectric transducers (PZT2 to PZT5), identical to PZT1, were placed near the corners of the plate 140 mm from the central piezoelectric disc and acted as reference receivers (see Figure 2b). The actuator and receiver PZTs were glued onto the plate surface using cyanoacrylate adhesive following a 200-mm square path around the central piezoelectric disc. Similarly, the sensing fiber was glued continuously following a 230-mm square path leaving a 6-m fiber loop between each pair of fiber strands. The choice of a 6-m loop was dictated by the necessity to spatially resolve the vibration signals collected by our ϕ-OTDR sensor along the four strands (the spatial resolution of the considered setup is 2 m).

The excitation signal applied to PZT1 for the generation of the Lamb wave was formed by a 5-cycle sine wave at 38 kHz with a Hamming window (see the blue curve in Figure 3). The signal was produced using an arbitrary waveform generator (AWG) (HP Agilent 33120A). The output of the AWG was connected to two linear amplifier modules (MX200 Piezo Drive) in order to amplify the voltage signal up to a peak-to-peak value of 400 V. The AWG was synchronized with the data acquisition system depicted in Figure 1, as well as with the oscilloscope used to capture the voltage waveforms from PZT2 to PZT5. In Figure 3, we show the excitation signal together with the signal detected by PZT2 over a time window of 1 µs. From this plot, the delay between the first peaks of the two waveforms, corresponding to the first symmetric Lamb mode S0, equals 25 μs. Considering that the distance between the two transducers was 140 mm, the resulting Lamb wave group velocity is equal to about 5600 m/s, which is in good agreement with theoretical and numerical expectations for the S0 mode [18]. It was also noticed that the first wave packet in the acquired waveform had more than five peaks. This was attributed to the overlap at the receiver of the S0 and A0 (antisymmetric) Lamb modes.

Regarding the fiber-optics measurements, an averaging strategy was developed to improve the signal-to-noise ratio, as schematically sketched in Figure 4. In brief, the method consists of performing and averaging 125 consecutive acquisitions at a trigger rate of 10 Hz (in Figure 4 this rate is represented by its period T_f_ = 100 ms). The excitation waveform was sent to PZT1 every 100 ms. The total time required to retrieve the whole dataset was then 125 × 100 ms = 12.5 s. The choice of a period T_f_ = 100 ms was made to allow the transient associated with the generation of Lamb waves to dissipate before the application of a new tone burst. For each tone burst sent to PZT1, the acoustic response of the fiber was recorded by performing 800 consecutive acquisitions of the ϕ-OTDR signal, with each acquisition being triggered by a 20-ns pulse (see Figure 4). These pulses were generated at a repetition rate of 900 kHz; therefore, the period $T_{s}$ shown in Figure 4 is equal to 1.11 μs. The choice of a pulse repetition rate of 900 kHz was dictated by the necessity to adequately sample the 38-kHz Lamb wave, while, at the same time, avoiding the overlap of the backscattered traces. According to the observations, the requirement of a high repetition rate in Lamb wave detection was found to limit the maximum length of the sensing fiber. For example, the 900-kHz acquisition rate limits the sensing length to just over 100 m. This is a limitation of the proposed method.

The acquisition process was accomplished by first recording the 125 × 800 = 100,000 ϕ-OTDR traces in the internal memory of the acquisition card, and then transferring the whole dataset to the PC connected to the acquisition card. After data transfer, the acquisitions were averaged to get an 800-sample acoustic signature for each fiber position.

Figure 5 shows the differential phase acquired using the ϕ-OTDR system and corresponding to the 200-mm portion of the fiber denoted with OFS1 in Figure 1 with the plate in rest conditions. The repeatability of the measurements in terms of waveform shape was verified by normalizing and superimposing three consecutive acquisitions. Taking the first acquisition as a reference, the norm of the difference between the two successive traces and the first trace, normalized to the norm of the first trace, was about 15%. Therefore, even if the overall shape is retained, some differences between consecutive acquisitions remain, probably due to some environmental instability affecting the retrieved optical phase.

It is also interesting to compare the ϕ-OTDR signal with the PZT signal acquired in a nearby position. Therefore, the time-aligned signals acquired by PZT2, PZT3, and OFS1 are shown in Figure 6a. Limiting the analysis to the first wave packet, it can be seen that the PZT and optical fiber signals show some similarities, but they are not identical. This can be explained by considering that, while the PZT signals are collected at the piezo transducer position, the optical signal is the dynamic phase shift cumulated over the whole fiber strand (OFS1, in this case) due to the limited (2 m) spatial resolution. Therefore, a perfect equivalence cannot be expected. However, comparing the signals in terms of their power spectral density (see Figure 6b), it can be seen that the signals share a significant spectral portion around the central frequency of the excited Lamb wave (38 kHz). It is also interesting to note that the signals also exhibit some spurious components at higher frequencies (e.g., around 120 kHz and 150 kHz). These spectral disturbances have been verified to be introduced into the excitation waveform by the PZT amplifier modules.

As a next step, the response of the four piezoelectric transducers PZT2-PTZ5, as well as that of the four fiber portions attached to the plate was analyzed, with the latter being in either a rest or a cantilever condition. The application of a static load to the plate was aimed only at inducing within the plate a stress-strain state altering Lamb wave propagation mechanisms, and not to analyze in detail nor compensate the effects of an applied load on Lamb wave propagation [18]. For the cantilevered case, the deformation of the plate due to its own weight was determined using the same optical fiber employed for the acoustic measurements by connecting its ends to a laboratory prototype implementing the Brillouin optical frequency-domain analysis (BOFDA) technique as described in Ref. [19]. The measurement, carried out at a spatial resolution of 16 mm and shown in Figure 7, revealed a maximum strain on OFS1 and OFS3 equal to about 160 µε. For both fibers, the maximum strain occurred near the fixed end of the plate, while decreasing linearly when moving towards the free end. Note that in Figure 7, the distance used for the horizontal scale is measured with respect to the fixed end of the plate. On the other hand, the strain measured along OFS2 and OFS4 by the same apparatus was negligibly small.

Then, we report in Figure 8, the response of the four piezoelectric sensors for the two plate conditions. It is interesting to evaluate the norm of the difference between the responses of each PZT when the plate is in a rested or cantilevered state. Table 1 summarizes the results relative to each PZT. Interestingly, the maximum variation occurred for PTZ4, which is one of the two transducers placed close to the fixed end of the plate, i.e., where strain is maximum.

Figure 9 shows the results of the optical fiber acoustic sensor. Again, the results obtained with the plate in rest conditions are compared to those obtained with the plate in the cantilever condition. Similarly to the procedure carried out for the PZT sensors, we have calculated the norm of the difference between the acquired signals. The results are reported in Table 2.

We observe that the highest variation is recorded along OFS4, i.e., the fiber strand placed near the free side of the plate. However, it should be pointed out that OFS4 was also the fiber strand recording the weaker signal in terms of induced phase shift. Regardless, the data highlight that the ϕ-OTDR signal is affected in all fiber strands by the modification of the plate condition to a larger extent compared to the piezo transducers. This higher sensitivity can be explained by considering the distributed nature of the optical sensing technology employed for our measurements. In fact, the signal recorded by each fiber strand glued along the plate is the result of the mechanical vibration of the plate integrated over the whole strand; therefore, it is highly sensitive to any perturbation affecting the propagation of the Lamb waves. We believe that the response of the optical fiber sensor would be much closer to that of PZT sensors after improving the spatial resolution of phase-OTDR measurements down to the cm-scale. It is also noticed that, while for the present case this higher sensitivity is somewhat detrimental, as the Lamb wave propagation mechanisms were altered by a static load, other cases of interest are those in which the perturbation is induced by some mechanical defects. These test cases will be the subject of future investigations.

While the values reported in Table 1 and Table 2 are global parameters which depend on the detected signals over the whole acquisition interval, it is also useful to focus on some specific parameters of the detected Lamb waves. In particular, attention was paid to the main peak of the first wave packet of the detected Lamb waves. In Table 3 and Table 4, the corresponding amplitude and position of this peak were summarized for the plate in rest and cantilever conditions and for both sensing technologies. In contrast to the norm values reported in Table 1 and Table 2, the values reported in Table 3 and Table 4 indicate that the first wave packet is much less sensitive to the load conditions of the plate. In particular, for the PZTs the only variation relates to the decrease in the peak amplitude of PZT5, putting the plate in a cantilevered condition. Similarly, the ϕ-OTDR measurements only reveal a decrease in the peak amplitude over the OFS1 and OFS3 fiber strands, which are the strands deployed along the direction of the strain gradient imposed by the static load.

## 4. Conclusions

In this work, the ϕ-OTDR technology has been used to detect the Lamb waves produced by a piezoelectric transducer in an aluminum plate. The experiments demonstrate the feasibility of the proposed methodology in distinguishing the Lamb waves in different portions of the fiber, as well as in determining the variations of the mechanical response of the plate resulting from a change in its load condition. While the tests reported in this paper have been conducted with only four sensing positions, the use of more advanced ϕ-OTDR configurations, with submeter [20] or even cm-scale [21] spatial resolution, may provide much more detailed information about the Lamb wave propagation in the investigated structure.

Further efforts are needed to verify the capability of the proposed method to identify and locate damage in the investigated structure, as well as to improve the spatial resolution and, therefore, the number of effective sensing points.

## Figures and Tables

**Figure 1 sensors-22-05962-f001:**
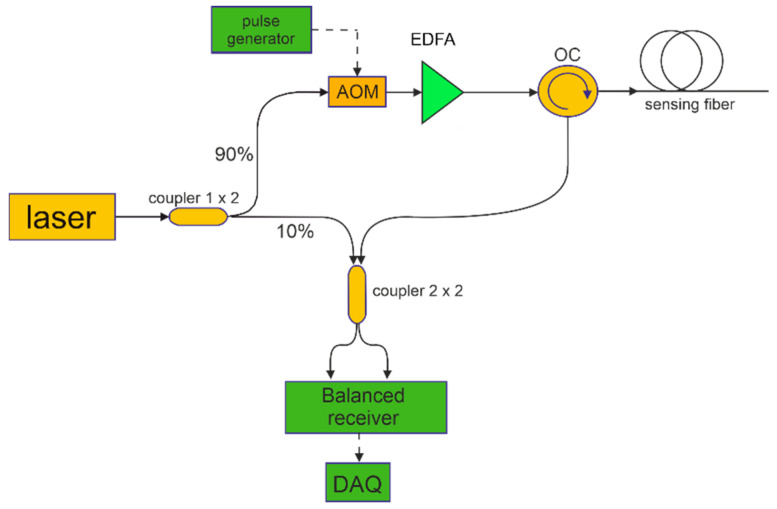
Experimental setup employed for Lamb wave detection based on the ϕ-OTDR method. AOM: acousto-optic modulator; EDFA: erbium-doped fiber amplifier; OC: optical circulator; DAQ: data acquisition.

**Figure 2 sensors-22-05962-f002:**
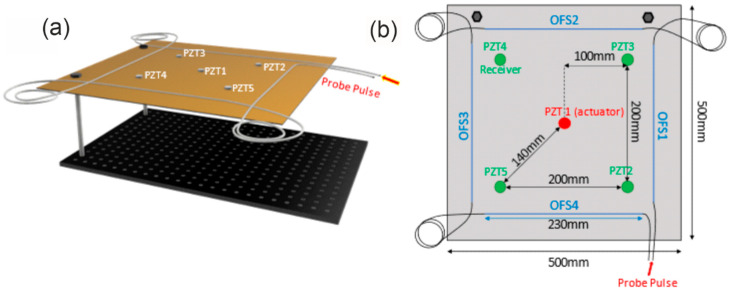
(**a**) Aluminum plate used for the experiments in the cantilever position. The PZT1 acted as an actuator, while PZT2, PZT3, PZT4, and PZT5 acted as receivers. The red arrow indicates the start of the FUT. (**b**) Details of the FUT and PZTs arrangement (upper view). The red circle represents the PZT acting as an actuator, while the green circles represent the PZTs acting as receivers. The blue lines represent the fiber strands glued along the plate.

**Figure 3 sensors-22-05962-f003:**
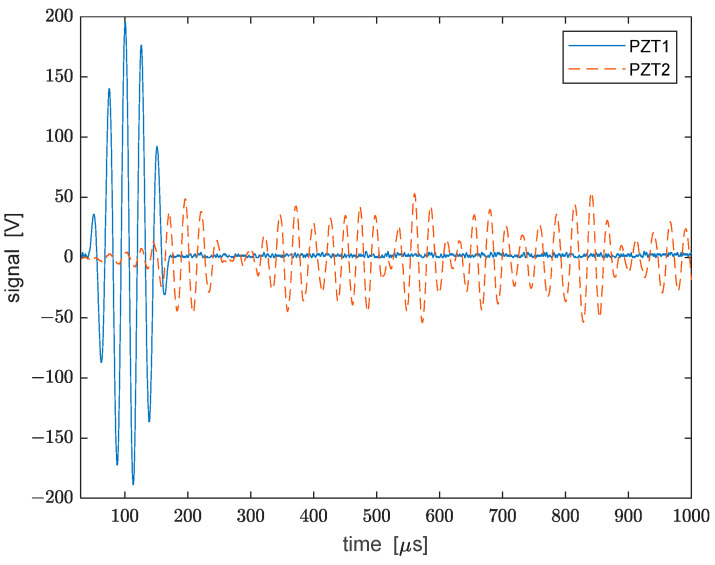
Excitation signal transmitted to PZT1 (blue solid line) and detected signal recorded by PZT2 (red dashed line). The detected signal from PZT2 has been multiplied by a factor of 20 for clarity.

**Figure 4 sensors-22-05962-f004:**
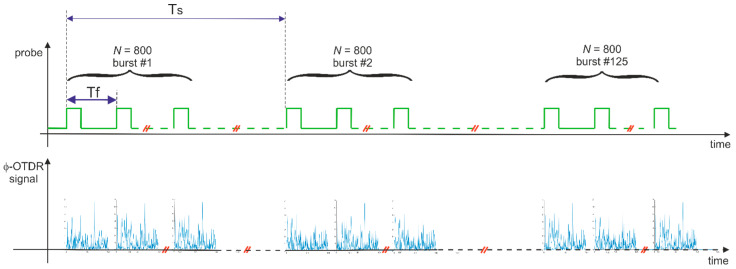
Time scales of the ϕ-OTDR measurements. $T_{f}$ represents the repetition of the pulses used to capture the acoustic response of the fiber, while $T_{S}\;$ represents the repetition period of the tone bursts used to excite the Lamb waves in the aluminum plate. The red diagonal lines along the horizontal axes represent time gaps.

**Figure 5 sensors-22-05962-f005:**
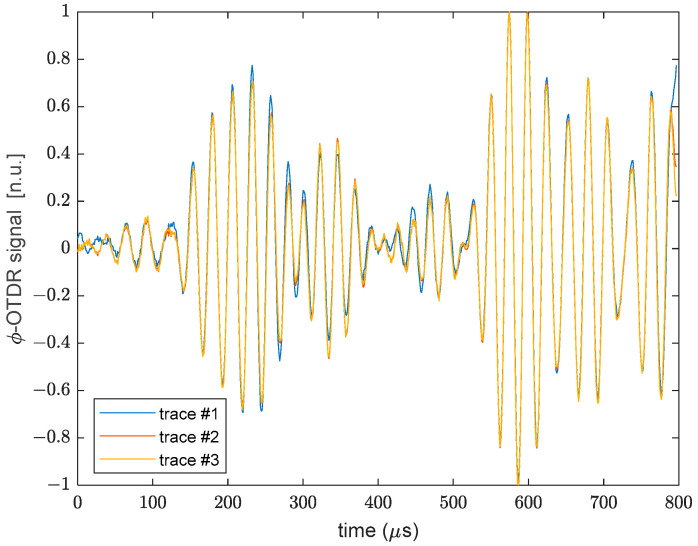
Normalized ϕ-OTDR acquired along the OFS1 fiber strand over three consecutive measurements.

**Figure 6 sensors-22-05962-f006:**
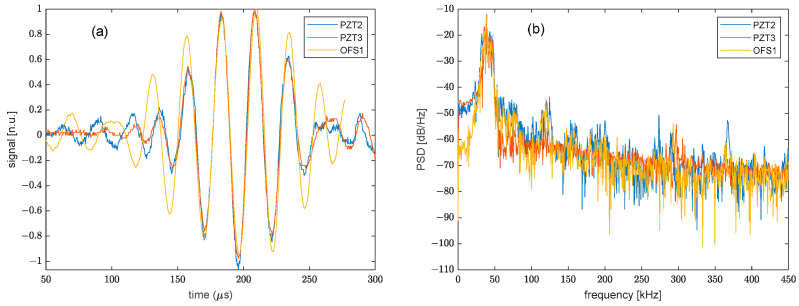
(**a**) Response of PZT2, PZT3, and OFS1, with the latter temporally translated for alignment with the PZT signals and (**b**) corresponding power spectral densities.

**Figure 7 sensors-22-05962-f007:**
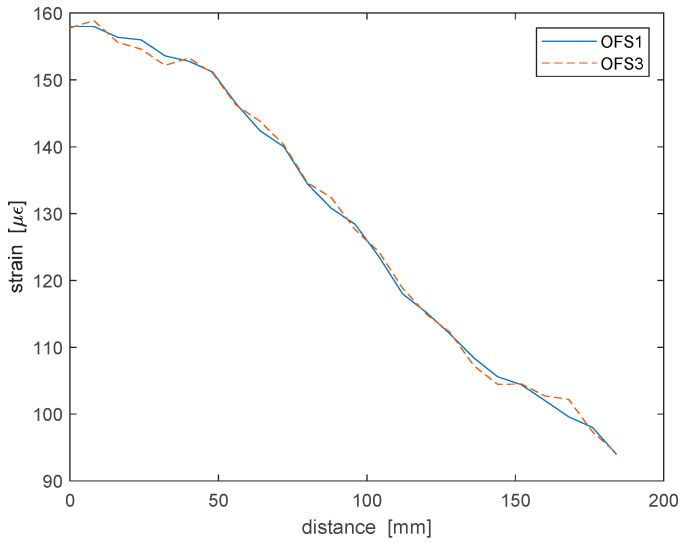
Distributed strain measurement along the optical fiber strands OFS1 and OFS3 with the plate in the cantilevered state.

**Figure 8 sensors-22-05962-f008:**
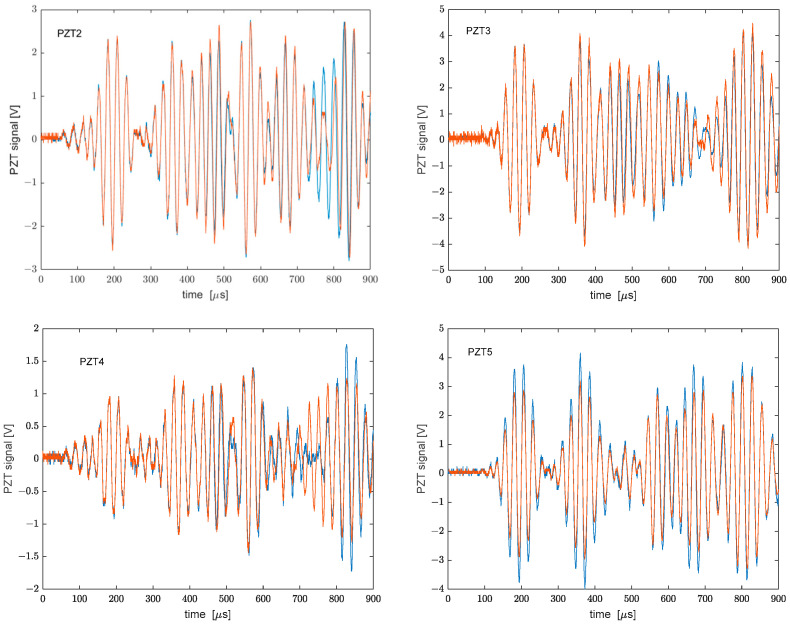
Responses of the PZTs with the plate in rest conditions (blue curves) or cantilever conditions (red curves).

**Figure 9 sensors-22-05962-f009:**
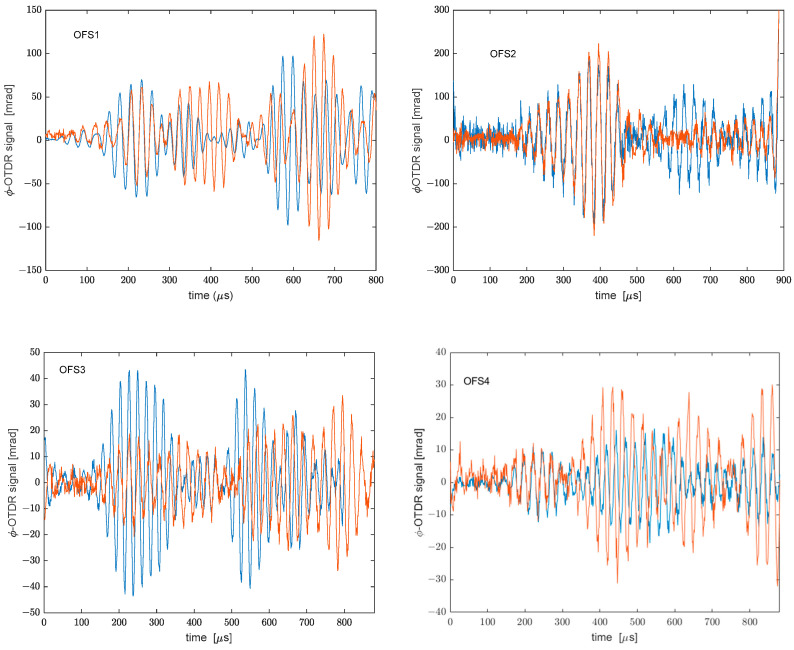
Responses of the ϕ-OTDR sensor with the plate in rest conditions (blue curves) or cantilever conditions (red curves).

**Table 1 sensors-22-05962-t001:** Norm of the difference between the PZT responses acquired with the plate in rest conditions or cantilever condition.

	PZT2	PZT3	PZT4	PZT5
Norm of the difference	31.84%	26.74%	41.13%	23.62%

**Table 2 sensors-22-05962-t002:** Norm of the difference between the ϕ-OTDR signals acquired with the plate in rest conditions or cantilever condition.

	OFS1	OFS2	OFS3	OFS4
Norm of the difference	115.23%	61.97%	108.12%	227.61%

**Table 3 sensors-22-05962-t003:** Main parameters of the Lamb wave detected by the PZTs.

Rest Conditions	PZT2	PZT3	PZT4	PZT5
Peak position [μs]	209	206	206	206
Peak amplitude [V]	2.3	3.6	0.9	3.7
Cantilevered conditions				
Peak position [μs]	209	206	206	206
Peak amplitude [V]	2.3	3.6	0.9	2.8

**Table 4 sensors-22-05962-t004:** Main parameters of the Lamb wave detected by the ϕ-OTDR sensor.

Rest Conditions	OFS1	OFS2	OFS3	OFS4
Peak position [μs]	232	232	227	222
Peak amplitude [mrad]	81	65	43	10
Cantilevered conditions				
Peak position [μs]	232	232	227	222
Peak amplitude [mrad]	60	65	17	10

## Data Availability

Data are available upon reasonable request.

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
