# Peer review of "Lamb Wave Detection for Structural Health Monitoring Using a ϕ-OTDR Systemâ€"

_sensors, 2022, doi:10.3390/s22165962_

Round 1
Reviewer 1 Report
The revision is ok. But the author should know that if your transduction mechanism will be applied for damage detection, then high sensitive to load may be a major drawback that will make your damage detection algorithm much complex than using PZT.
Therefore, your mechanism may be applied at a different scenario other than SHM of damage detection.
Author Response
We thank the Reviewer for his/her comment. We agree with the Reviewer that a higher sensitivity to the load conditions may be a drawback when the system is used for damage detection. This has been also underlined in the manuscript (lines 281-285). However, as discussed in the revised paper (page 10), restricting the analysis to the main peak of the first wavepacket of the detected Lamb wave, both PZTs and optical fiber measurements are much less sensitive to the load conditions of the plate. In particular, the phase-OTDR measurements only reveal a decrease of the peak amplitude over the OFS1 and OFS3 fiber strands, which are the strands deployed along the direction of the strain gradient imposed by the static load.
Reviewer 2 Report
The current manuscript is a revised version of the previously submitted one. The authors have somehow addressed most of the reviewers’ concerns. I still need to point out that the authors only prove that the sensitivity of using phase-OTDR system is higher than that of the conventional PZT sensors in terms of detecting the variations of Lamb wave when there is a static load. However, there are no quantitative measurements or analysis regarding how much improvement in sensitivity is obtained when using the fiber-optics sensing technique, which makes the work a superficial one. Would the higher sensitivity be due to the deformation of the fiber strands in the phase-OTDR system but not the sensing technique itself or due to the alternation of the propagating Lamb wave after the plate being deformed? Analysis should be given for this issue as it can be seen from Fig.9 the phase of the detected Lamb wave is altered after the plate is deformed while the phase remains quite stable in Fig.8. Meanwhile, the authors still do not convince me why the phase-OTDR system should be used in this work rather than FBG sensors as the distributed sensing capability of the sensor is not fully exploited.
Author Response
1) We thank the Reviewer for his/her comments. As explained in our previous reply, phase-OTDR sensors only detect the dynamic strain, as the perturbation in each fiber position is determined by comparing the phase of the backscatter signals produced by consecutive pulses. Therefore, the variations of signals in Fig. 9 represent the modification of the Lamb wave produced by the static load. The quantitative analysis is reported in Tables I to IV. As regards the comparison with Fig. 8 (PZT response), we believe that the higher sensitivity of phase-OTDR measurements should be attributed to the distributed sensing nature of this technology. As discussed in page 10, the signal recorded by each fiber strand glued along the plate is the result of the mechanical vibration of the plate integrated over the whole strand, therefore it is strongly sensitive to any perturbation affecting the propagation of the Lamb waves. We believe that the response of the optical fiber sensor would be much closer to that of PZT sensors, after improving the spatial resolution of phase-OTDR measurements down to the cm-scale. This point has been added in the revised manuscript (lines 279-281).
2) As discussed in our previous reply, the phase-OTDR system used in our tests was able to collect the dynamic strain signal from four positions of the plate. We agree with the Reviewer that, having only four sensing positions does not permit to fully take advantage of the proposed sensing technology. This was a direct consequence of the limited spatial resolution (2-m) available for our tests. Enhancing the spatial resolution down to the cm-range (which is challenging, but technically feasible), would increase notably the number of sensing points, and therefore the attractiveness of the proposed method. We should also underline that the results presented here are the first demonstration of Lamb wave detection using a phase-OTDR sensor. We hope that these preliminary results will stimulate further experiments in the field of distributed acoustic sensing for Lamb-wave based SHM.
This manuscript is a resubmission of an earlier submission. The following is a list of the peer review reports and author responses from that submission.
Round 1
Reviewer 1 Report
In the manuscript, the authors employed a phase sensitive optical time domain reflectometry to measure the Lamb wave in an aluminum plate. The working principle is well explained and the measurement result of the φOTDR agrees with other sensors. I think the manuscript can be published if the following points can be addressed:
1. Line 67, the phase change is not a local event. Actually a phase delay is introduced due to the perturbation, so the phase of the light backscattered after the perturbated section changes.
2. Line 130, “P” is missing for Piezo-ceramic.
3. Please make it clear, whether all the sensing fiber is glued or just few points.
4. The pulse repetition rate of the φOTDR should be very high to sample the Lamb wave, which greatly limits the sensing distance of the system. The authors need to mention this “drawback” in the paper to provide the reader a full picture.
Reviewer 2 Report
The author uses a phase-sensitive optical time-domain reflectometry sensor for Lamb wave detection in an aluminum plate. The system can detect and resolve the Lamb waves generated by a piezoelectric transducer in distinct regions of the plate, making it possible for realizing SHM using a single optical fiber attached to the structure. I recommend rejection of the paper with the following comments:
1. Languages can be modified by a native speaker. Please make language concise.
2. How do you know the guided wave signals you collected from the optical sensor are correct? Since the traditional PZT receiver detect the guided wave signal in a single point, please specify which location of the plate the optical sensor is receiving the guided wave. And at the very same location, put a PZT receiver to obtain the guided wave signal. And then make comparison so that it can prove if your sensor is getting the right signal. Or you can perform a FEA simulation to theoretically obtain the signal at the specified location of the plate, then make comparison.
3. The experiment design in this study is questionable. It is not fair to say that the optical sensor is more sensitive than conventional PZT receiver because you are essentially mixing two concepts. We all know that fiber optics are sensitive to strains. Therefore, the larger variation can result directly from the load effect to the optical sensor, nothing related to the guided wave propagation. In fact, ideally, we want the guided wave propagation to be invariant to load effect in order to detect damage in a structure. You test results are actually proving that PZTs are better than the optical sensor in terms of the resistance to load effect for guided wave propagations.
Reviewer 3 Report
In this paper, the phase-OTDR technique is used to detect the Lamb waves generated by piezoelectric transducer in an aluminum plate. It is shown that the proposed methodology is able to distinguish Lamb waves in different portions of the fiber as well as the mechanical response of the aluminum plate. Though the authors do show some experimental results, I didn’t see any useful info that is revealed by the proposed technique concerning the distributed sensing capability of the phase-OTDR system and the detected Lamb waves as well as the deformation of the plate. In my point of view, the manuscript cannot be accepted in its current form with the following concerns:
1. The authors use the detected results of the Norm of the difference between the two plate conditions to distinguish the conditions of the plate and the positions of the highest variation in the detected Lamb wave. However, the authors failed in defining an accurate standard concerning the difference values in determining the lamb wave variations. The usage of only the largest difference value is not accurate and precise as the measurement results would be disturb by other environmental noises when using phase-OTDR system.
2. Without showing the distributed measurement ability using the phase-OTDR system, it is really doubtful why the authors use phase-OTDR system for the measurements. It would be very costly using phase-OTDR system in the current case without utilizing its distributed sensing capability. FBG sensors would be more suitable for the issue raised by the authors in the current manuscript.
3. I didn’t see any useful detection info concerning the detected Lamb waves, including their frequencies, amplitudes, positions and so on. The authors should provide these info to improve the quality of their work.